# Methodology to Prioritize Chilean Tailings Selection, According to Their Potential Risks

**DOI:** 10.3390/ijerph17113948

**Published:** 2020-06-02

**Authors:** Elizabeth J. Lam, Italo L. Montofré, Fernando A. Álvarez, Natalia F. Gaete, Diego A. Poblete, Rodrigo J. Rojas

**Affiliations:** 1Chemical Engineering Department, Universidad Católica del Norte, Antofagasta CP 1270709, Chile; natalia.gaete.r@gmail.com (N.F.G.); diego.poblete@ucn.cl (D.A.P.); rrojas02@ucn.cl (R.J.R.); 2Mining Business School, ENM, Universidad Católica del Norte, Antofagasta CP 1270709, Chile; falvarez@ucn.cl; 3Mining and Metallurgical Engineering Department, Universidad Católica del Norte, Antofagasta CP 1270709, Chile; 4Administration Department, Universidad Católica del Norte, Antofagasta CP 1270709, Chile

**Keywords:** mine tailings, environmental liabilities, mining industries, environmental risks

## Abstract

For centuries, Chile has been a territory with significant mining activity, resulting in associated social benefits and impacts. One of the main challenges the country faces today is the presence of a great number of mine tailings containing heavy metals, such as Cu, Cr, Ni, Zn, Pb, As, Cd, and Fe, which make up a potential risk for the population. This study is intended to develop a methodology for determining tailings requiring urgent treatment in Chile, based on risks associated with heavy metals. Geochemical data from 530 Chilean tailings were compared to the Dutch norm and the Canadian and Australian soil quality guidelines for residential use. Additionally, criteria about residents and water bodies were used, considering a 2-km area of influence around tailings. To do this, QGIS (Böschacherstrasse 10a CH-8624 Grüt (Gossau ZH), Zurich, Switzerland), a geospatial tool, was used to geolocate each deposit, considering regions, communes, rivers, lakes, and populated areas. To evaluate potential ecological contamination risks, Hakanson’s methodology was used. Results revealed the presence of 12 critical tailings in Chile that require urgent treatment. From the 530 tailings evaluated, 195 are located at less than 2 km from a populated area and 154 at less than 2 km from a water body. In addition, 347 deposits require intervention: 30 on Cu, 30 on Cr, 13 on Zn, 69 on Pb, 138 on As, 1 on Cd, and 5 on Hg.

## 1. Introduction

Chile is recognized for exploiting its mineral resources [1] on a world basis, a fact that has brought about economic benefits, but at the same time, a negative impact on the environment, one of the most severe being the generation of mining environmental liabilities (MEL) [2,3]. These are distributed along the country; some of them are abandoned and without due management and maintenance. According to a survey made in 2017 by the National Geology and Mining Service (SERNAGEOMIN, for its acronym in Spanish) [4], there are 740 tailings in Chile: 14.1% of them are classified as active, 62.6% as inactive (non-abandoned), and 23.3% as abandoned (inactive). From the three groups, the latter requires the most attention, their management being still uncertain since they are not thoroughly characterized. For this reason, three management alternatives are considered: (1) Recovering metals of interest, mainly from old deposits with high mineral grades, which could be further exploited since current technology can help extract more minerals from them [5,6,7]; (2) utilizing technological solutions for reusing tailings, e.g., as construction aggregates [8]; (3) tailings disposal in a stable manner [9]. 

A high percentage of tailings are old [10]. Their construction was not subjected to state supervision owing to the lack of regulations, thus representing an even greater risk [9]. In the last decade, although Chile has gradually increased its mining regulatory framework to benefit the environment, there are still serious weaknesses. The Chilean territory is characterized by different types of climates and soils [11], thus making background metal concentrations quite different from each other in the different regions. On a national basis, there is a scarce amount of background metal concentrations. This has not allowed the establishment of regulations to assess the risks that the concentration of certain substances in the soil could imply. So, international regulations are used, resulting from the natural soil conditions of foreign countries. Considering that Chile shows a geological soil diversity and a natural mineralogical abundance, the uncertainty concerning the use of these adopted regulations is high. In 2012, the Chilean Ministry of Environment approved the Methodological Guide for Managing Soils with Potential Presence of Contaminants (GMPGSPPC, for its acronym in Spanish) [12].

In developed countries such as The Netherlands, Canada, and Australia, the establishment of reference values for heavy metals in soils has allowed the improvement of their planning and environmental management, thus becoming a control device for environmental entities. Since Chile does not count on these reference values, although it does have a large amount of soils with a potential presence of contaminants, mainly from mine tailings, the country needs a methodology to estimate the priority to intervene in the sites exposed to them.

Risks associated with tailings are critical because the latter are large masses that could suffer physical and/or chemical destabilizations, that is, the collapse or overflow of the solid materials in the dam, along with chemical reactions resulting in acid mining drainage [13,14,15]. Thus, tools to create a record of tailings requiring intervention is urgently needed. Health risks associated with mineral exposures are widely reported in the literature. Particularly, early exposure to tailings has been associated with the prevalence of congenital malformations, diarrhea, asthma, endocrine and neurological disorders, among others, neurodevelopmental delays and deficits, obesity, diabetes, and cancer—some of which have been increasing over the last few decades [16,17].

The objective of the study is to estimate whether Chilean tailings involve contamination risks. To do this, an applied methodology combining the use of algebraic equations and international reference values with environmental and demographic criteria related to the environmental risk stated in GMPGSPPC is presented. Finally, an assessment of the potential ecological risk resulting from heavy metal contamination is made by using Hakanson’s methodology [18], which was chosen because it is one of the methods widely used to assess the potential ecological risks of mining soils [19,20,21,22,23,24,25]. This methodology allowed the creation of a record of Chilean tailings to be prioritized for urgent intervention, control, and management. 

## 2. Materials and Methods

### 2.1. Experimental Data

Experimental data were obtained from the geochemical database of the Chilean Geochemical Characterization Program for tailings from SERNAGEOMIN [4]. The data corresponded to 631 out of the 740 tailings registered in the country. The characterization involved the measurement of concentrations in g ton^−1^ of 56 elements and species per sample, considering potential risks and economic values associated with elements of value. Other data available in the database were the type of mining originating the tailings, the state of the tailings (active, inactive, abandoned), their location (region, commune, and geographic coordinates), and mass in estimated tons. These data are available only for a small number of deposits.

Eight elements and species characterized in the sample were evaluated: As, Cd, Cr, Cu, Fe, Ni, Pb, and Zn. In this study, the geochemical data identification number (IDQ), provided by the SERNAGEOMIN database aforementioned, was used for each measurement to identify tailings. Only 530 from the 631 dams had georeferentiation data, so 530 dams were assessed.

### 2.2. Tailings Characteristics

From the 530 tailings, 84 were classified as active, 421 as inactive, and 25 as abandoned. On a regional basis, most tailings were found in the Coquimbo region (62%), the Atacama region (19%), and the Antofagasta region (6%).

According to the type of mining originating them, the following tailings were found: Ag-Au (2), Au (66), Au-Cu (15), Au-Cu-Ag (11), Au-Zn (2), Cu (193), Cu-Au (99), Cu-Mo (13), Cu-Au-Ag(1), Cu-Au-Fe (1), Fe (4), Zn(2), Zn-Cu (2), limestone (2), sediment (96), unknown origin (8), and extremely old dams or stockpiles (2).

### 2.3. Methodology

Figure 1 shows the methodology proposed, which consists of three phases. Phase 1: Identifying tailings exceeding the admissibility of the reference concentration values of As, Cd, Cr, Cu, Fe, Ni, Pb, and Zn from the Canadian [26] and Australian [27] guidelines and the Dutch norm [28]. These countries were chosen for the following reasons: (1) The Netherlands is considered as the country with the most experience and development for protecting soil contamination, their norm being the most used in different parts of the world that lack reference values [29,30,31]; (2) Canadian standard values have been used as a reference for soil quality in Chile [9,12]; (3) Australia has climatic and geological characteristics quite similar to Chile [32]. This phase provided a record of tailings whose heavy metal concentration exceeded the values of the three international reference guidelines. 

Phase 2: The tailings identified in Phase 1 were assessed according to two of the priority criteria established in GMPGSPPC [12]. This phase allowed us to obtain a record of the tailings representing a higher risk for the health and safety of the population around them. A subrecord of the tailings identified in Phase 1 was obtained, also including criteria considered as a priority for Chile. Phase 3: The potential risk of deposits showing heavy metal concentrations over reference values, which were also considered as critical in Phase 2, according to the evaluation of risk for health and safety, was estimated by using the methodology by Hakanson [18], who defined mathematical equations and parameters to estimate contaminant-potential and the ecological risk of different heavy metals. The results allowed us to obtain a subrecord from the one obtained in Phase 2, which identifies Chilean tailings requiring urgent intervention.

The methodology followed for each phase shown in Figure 1 is described below.

#### 2.3.1. Phase 1: Comparison of Heavy Metal Concentrations, Using International Reference Values

##### Reference Frameworks

The geochemical data of tailings were compared with the standard values of soils used by the population, established by the Canada Soil Quality Guidelines [26], the Australian guidelines [27], and the standards established by the Ministry of Infrastructure and Water Management (MIWM) in the Dutch Soil Regulation Circular [28].

Canadian and Australia Guidelines

Given the great variability of reference concentration values provided by the Canadian and Australian guidelines, the first parametrization was made by using the equation by Esquenazi et al. [9]:(1)$$C^{*}=\mathrm{sign}\left(C\right)\cdot \log\left[1+\mathrm{abs}\;\left(C\right)\right]$$
where C is the concentration of the element or substance, and C* is the corresponding parameterized value. The Canadian and Australian parameterized values for industrial use are shown in Table 1. The experimental values of heavy metal concentrations corresponding to the 530 tailings assessed were also parameterized with Equation (1). Then, they were compared with the reference values in Table 1.

Dutch norm

The Dutch norm is based on an algebraic formula that allows the adaptation of its use, depending on soil nature. Its parameters include the standard intervention value (SIV), which depends on two parameters characteristic of the soil: the percentage of organic material weight and the percentage of clay weight. Although this norm does not have a legal value in Chile, it has been the reference most frequently used by SERNAGEOMIN so far, the same as in other countries [33,34,35,36]. Since the SERNAGEOMIN geochemical database does not include the measurement of clay percentage nor the percentage of organic material, the Dutch norm adapted to mine tailings by Esquenazi et al. [9] was used. This methodology allows the definition of risk zones for the population and/or the environment by means of three criteria: (1) intervention required, (2) conditional intervention required, and (3) no intervention required [9].

The soil intervention value (SIV) is defined by Equation (2):(2)$$\mathrm{SIV}=\mathrm{SSIV}\;\cdot \;\frac{A+B\;\cdot {\;x}_{C}+\;B\;\cdot x_{\mathrm{OM}}}{A+25\;\cdot B+10\;\cdot C}$$
where A, B, and C are the specific parameters of each metal, and $x_{\mathrm{OM}}$ and $x_{C}$ are the percentage values of organic material and clay weight, respectively. Table 2 shows the values of A, B, and C for the eight heavy metals to be assessed, assuming 25% clay and 10% organic material. SSIV is the intervention value of a standard soil for residential use, having the following values: As = 27 mg kg^−1^; Cd = 1.2 mg kg^−1^; Hg = 0.83 mg kg^−1^; Pb = 2.10 mg kg^−1^; Co = 35 mg kg^−1^; Cu = 54 mg kg^−1^; Ni = 34 mg kg^−1^ and Zn = 200 mg kg^−1^ [28].

Adjusted threshold values were estimated for each heavy metal ($C_{F}$) by applying Equation (3) [9,37].
(3)$$C_{F}=\frac{\mathrm{SIV}}{\mathrm{SSIV}}\cdot \left(\frac{A+25\cdot B+10\cdot C}{B}\right)-\frac{\left(A+2C\right)}{B}$$

To simplify the analysis, $C_{F}$ values were parametrized according to the equation by Esquenazi et al. [9], defined as
(4)$${\mathrm{AC}}_{F}=\mathrm{sign}\left(C_{F}\right)\cdot \log\left(1+\mathrm{abs}\left(C_{F}\right)\right)$$
If ${\mathrm{AC}}_{F}<0$, no intervention required.If ${\mathrm{AC}}_{F}>2$, intervention required.If $0<{\mathrm{AC}}_{F}<2$, intervention depends on the values of other parameters.${\mathrm{AC}}_{F}$ are the reference values of the Dutch norm.

##### Intervention Requirements Due to Contamination

The heavy metal concentration values of the 530 tailings under assessment were parameterized by using Equation (1). These were compared with the Canadian and Australian reference values (Table 1) and the values obtained from the Dutch norm equations (Equations (2)–(4)).

As a result of the three referential frameworks, the “no intervention required” and “intervention required” criteria were obtained. Additionally, in the case of the adapted Dutch methodology, the “conditional intervention” criterion was obtained. Its conditionality is subjected to the availability of the percentage values of organic material and clay in the tailings. These data are not available from the SERNAGEOMIN geochemical database.

The selection of deposits requiring urgent intervention depends on how their concentrations exceed the admissibility indicated by the Canadian and Australian guidelines and the Dutch norm. In this study, the number of metals classified as “intervention required” by the three international reference values was established as an indicator of the initial tailings prioritization. The priority criteria determined in this study were the following: low priority (0–2 metals requiring intervention), medium priority (3–4 metals requiring intervention), and high priority (5 or more metals requiring intervention).

#### 2.3.2. Phase 2: Prioritization Criteria, According to the Methodological Guidelines for Soils with the Potential Presence of Contaminants

Once the tailings were classified according to the intervention prioritization described above, a geospatial tool, QGIS, was used and added to the geographic data layers present in Chilean databases to determine their location. To prioritize soils with potential presence of contaminants (SPPC), two out the four criteria indicated by the methodological guidelines approved by the Chilean Ministry of Health to manage them were used [10]. The objective of these guidelines is to prioritize SPPC, as follows:

Distance from tailings to a populated area: less than 2 km—“high priority”, 2–3 km—“medium priority”, and greater than 3 km—“low priority”.

Closeness to water bodies: If tailings are close to water bodies, they are considered as “high priority”. If not, they are considered as “low priority”.

Table 3 shows the criteria for classifying Chilean tailings priority. The prioritization criterion “extreme” was applied to tailings with two or more red boxes, without disregarding the case of one red box, which also requires “high priority”. 

#### 2.3.3. Phase 3: Potential Risk Estimation

Lastly, the potential environmental risk of tailings identified as the most critical in the previous phases was estimated. To estimate potential tailings, Hakanson’s methodology [18] was used. He defined mathematical equations and parameters to estimate potential contaminants and the ecological risk factors of various heavy metals. Hakanson proposed the “Potential Ecological Risk Index” (PERI) as a quick tool for environmental assessment, resulting in the classification of contamination areas and the identification of toxic substances of interest. PERI provides a simple quantitative method for assessing the ecological risk potential of a contamination situation [38]. The equations used are the following [39]:(5)$$C_{f}^{i}=C^{i}/C_{n}^{i}$$
where $C_{f}^{i}$ is the contaminant factor of heavy metals “i”, $C^{i}$ is heavy metal concentration, and $C_{n}^{i}$ represents the heavy metal concentration in a nearby area without anthropogenic intervention (background level). $E_{r}^{i}$ is the ecological risk factor defined as
(6)$$E_{r}^{i}=T_{r}^{i}{*C}_{f}^{i}$$

$T_{r}^{i}$ is the heavy metal toxicity factor. The values of the heavy metal toxicity factors, $T_{r}^{i}$, were Hg (40), Cd (30), As (10), Cu (5), Pb (5), Cr (2), Zn (1), and Ni (6) [14,40].

On the basis of contaminant and ecological risk factors, a potential ecological risk index, RI, was determined through the following addition:(7)$$\mathrm{RI}=\overset{n}{\underset{i=1}{\sum }}E_{r}^{i}$$

Equation (7) allows us to estimate the potential ecological risk, considering the criteria in Table 4. [41].

## 3. Results and Discussion

### 3.1. Comparison of Intervention Requirements

There is a great discrepancy between the criteria of the three references, although all of them consider the soil for “residential” use, thus showing the need to have soil background values. These are scarcely measured in Chile, being a basic disadvantage for developing soil norms in the country. 

Table 5 shows the intervention requirements for the 530 tailings assessed, according to the three reference frameworks. Owing to the lack of data about the mass percentage of organic material and clay, there is a great number of tailings classified as “uncertain” (conditional intervention) according to the Dutch norm, thus making assessment difficult. Metals most frequently presenting this characteristic were Cr, Cd, and Ni, the percentage of “uncertain” being 73.4%, 77.9%, and 63.0%, respectively. The metals from the dams classified as “uncertain”, showing the smallest number of “uncertainties”, were Cu, Hg, Hg, and As, with 11.1%, 10.8%, 13.0%, and 13.0%, respectively.

Table 5 also shows that Cu is the metal requiring the most intervention, according to the three reference guidelines. Based on the reference values of the Dutch, Canadian, and Australian soils, 86.8%, 98.3%, and 65.5% of the dams require intervention, respectively.

The Australian reference values are the most demanding for As, Cd, and Hg, all of them being cancerogenic toxic substances, as shown by the great number of tailings requiring intervention due to the presence of these metals. Since Chile and Australia share similar climates, types of soil, and mining, these values must be considered.

A comparison was made among the results obtained for the three guidelines. Coincidences resulting from “intervention required” and “no intervention required” for the heavy metals assessed are shown in Figure 2.

In comparing requirements, there was a good agreement between The Netherlands and Australia (87.0%), while the agreement between The Netherlands and Canada amounted to only 26.0% for this metal. Concerning Cu, 67.2% agreement was found between the three guidelines, while the agreement between The Netherlands and Canada reached 88.5%. As to Cr, Cd, and Ni, the three guidelines show significant discrepancies, Cd being the most remarkable because only 1.1% of the 530 tailings agree with this criterion. The greatest agreement was found for Hg; that is, 85.7% of the criteria coincided for the three reference frameworks. A good agreement was also found for Pb and Zn; that is, 64.9% and 63.8%, respectively.

Table 5 shows a great number of “uncertain” criteria concerning the Dutch norm, thus decreasing the number of comparisons. In the case of Cr, Cd, and Ni, there are 73.4%, 77.9%, and 63.0% tailings that fit into the category of “uncertain” for these metals, respectively, making the application of the proposed methodology difficult.

In addition, the number of tailings requiring intervention was analyzed for the number of heavy metals exceeding maximum values, according to the three guidelines. As a result, 304 tailings required the intervention of one heavy metal; 117 tailings, two heavy metals; 77 tailings, three heavy metals; 28 tailings, four heavy metals; 4 tailings, five heavy metals, while none of the tailings required the intervention of six or more heavy metals.

The four tailings exceeding the three norms by five heavy metals are considered as “high priority”. They were identified by their IDQ geochemical data numbers. One of them is abandoned, while the other three are inactive. All of them contain a number of heavy metals exceeding the reference values of the three guidelines under assessment, that is, five heavy metals. The tailings size ranges from 600 to 1,875,000 t. Three of them are located in Sierra Gorda commune (Antofagasta Region) and one in Río Hurtado commune (Coquimbo Region). Table 6 shows their characteristics.

In applying the priority criteria mentioned above to the 530 tailings, 32 of them were classified as “high priority”; 195 as “medium priority”; and 303 as “low priority” for heavy metals. Tailings presenting the highest risks are located in Coquimbo, Atacama, and Antofagasta regions, an obvious fact since the country’s largest mining activity is concentrated in the Chilean northern zone. Figure 3 shows the tailings’ regional distribution.

### 3.2. Chilean Methodological Guideline Application

To apply the Chilean guidelines, it is first necessary to establish the location of each tailing. To do this, QGIS was used to geolocate them. This was compared with the layers available in the country, considering regions, communes, rivers, lakes, and populated areas. 

For geospatial data, a 2-km area of influence was created around the tailings based on data from Census 2012 (Chile, 2012) for the different locations. Figure 4 shows a Coquimbo region map as an example to indicate tailings and populated areas. Figure 4 shows the tailings area of influence of about 2 km around populated areas.

An analysis of the 530 tailings revealed that 195 and 290 of them are located at 2 and 3 km from a populated area, respectively. In addition, there are 154 tailings located near rivers, lakes, and estuaries. Their distribution is shown in Figure 5.

The assessment showed that there are 32 tailings classified as “high priority” due to heavy metal concentration, 195 due to closeness to populated areas, and 154 for closeness to water bodies. Figure 6 shows the results of the 530 tailings assessed, according to the number of heavy metals exceeding maximum values and closeness to populated areas by region. Figure 7 shows the results considering heavy metals and closeness to water bodies. Figure 6 shows tailings classified as “extreme”; that is, they present two critical characteristics: (1) the number of heavy metals exceeding maximum reference values is 5 or more, while (2) closeness to populated areas is smaller than 2 km. On the contrary, Figure 7 does not show any “extreme” tailings, indicating that, at the most, the heavy metals exceeding maximum values are 4.

### 3.3. Tailings Prioritized as “Extreme”

By considering heavy metal concentrations and closeness to populated areas and water bodies, tailings requiring special treatment were determined according to the following: (a) They show a high level of heavy metal contamination, (b) they are located within a 2-km radius close to a populated area, and/or (c) they are located within a water body area of influence. Twelve tailings classified as “extreme” were found, all of them located in the Illapel commune (Coquimbo Region). Five heavy metals exceed reference concentration values and are located at less than 2 km from populated areas and water bodies.

### 3.4. Ecological Risk Analysis

Hakanson’s methodology [14] was used to determine metals that are environmentally risky. To do this, soil background values are needed (Cu 134.7 g t^−1^; Cr 20.66 g t^−1^; Ni 50.68 g t^−1^; Zn 99.27 g t^−1^; Pb 7.881 g t^−1^; As 2.30 g t^−1^; Cd 1.508 g t^−1^; Hg 0.14 g t^−1^). The analysis using Hakanson’s methodology revealed that the five deposits identified as Anta Colla 1, Anta Colla 2, California 0, Esperanza Dos, and NN3 show high potential ecological risk indexes (RIs) greater than 320 and, therefore, are qualified as “showing extremely high ecological risk”, according to the criteria in Table 3. Figure 8 shows the location of four of them.

As shown in Figure 8, California 0 and Anta Colla 2 deposits are located close to populated areas. Hence, exposure to them cannot be controlled because their location makes it impossible to keep people away from the sites, the same for NN3 and Anta Colla 1 tailings located in the area. According to Illapel climatological data, rainfall is very low in summer months, while in winter (June and July), the average rainfall is 50 mm. Rainfall is a risk because metals may infiltrate the soil and contaminate nearby or underground water if the soil is permeable, and there is no appropriate insulation. Soils on the lower area of the Choapa River basin are mainly characterized by highly permeable soils and limited capacity for agricultural use, while in the upper area, the soils have characteristics appropriate for culturing fruit trees. The city of Illapel is located at upper Choapa, so its soil is less permeable.

Table 7 shows the potential ecological risk index values and the ecological risk factor of each heavy metal per tailings, revealing that the heavy metal with the highest ecological risk factor is As, the lowest being Zn, considering the average of the 12 critical deposits.

According to Hakanson’s ecological risk factor, the level of contamination was low for three of the twelve tailings assessed, that is, Cr, Ni, and Zn. In Anta Colla 2, the level of contamination was medium for Cu, low for Hg, very high for Cd, and extremely high for Pb and As, the latter being recognized for their high toxicity. In Anta Colla 1, the level of contamination was medium for Cd, high for Cu and Hg, and extremely high for Pb and As, the same as above. In California 0, the level of contamination was high for Cu and Cd, very high for Pb, and extremely high for As and Hg. In California 2B, the level of contamination was low for Cu, Pb, and Hg, medium for As, and very high for Cd. In El Arenal, San Jorge 1-2-3, and Tailings Dams 0, 1, and 2, the level of contamination was medium for As and Cu. In NN3 and Pluma de Oro, the level of contamination was low for Cu and medium for As. In Esperanza Dos, the level of contamination was high for Cu and medium for As. In Esperanza Dos and NN3, the level of contamination was extremely high for Hg, while in El Arenal, San Jorge 1-2-3, and Tailings Dams 1 and 2, the level of contamination was medium for Hg. In California 2B, Esperanza Dos, NN3, and San Jorge 1-2-3, the level of contamination was very high for Cd. In El Arenal, Pluma de Oro, and Tailings Dams 0, 1, and 2, the level of contamination was high. Five tailings fit into the category of “extremely high potential ecological risk”, that is, Anta Colla 1, Anta Colla 2, California 0, Esperanza Dos, and NN3. The other seven tailings fit into the category of “very high potential ecological risk”. These results indicate that the 12 tailings require intervention for their ecological and environmental risks. However, the order of priority is Anta Colla 2, Anta Colla 1, California 0, NN3, and Esperanza Dos.

## 4. Conclusions

This paper deals with a methodology to prioritize Chilean tailings according to their potential ecological and environmental risks. The application of this methodology shows that five tailings require urgent intervention in Chile; the heavy metals showing the greatest environmental risk being As, Cd, Pb, and Hg, which are recognized for their high toxicity.

The comparison of the three reference frameworks showed diverse criteria, thus making the analysis difficult. Only the tailings that do not require intervention, according to the Dutch, Canadian, and Australian reference frameworks, were discarded because they do not involve potential ecological and environmental risks. This allowed us to reduce the costs associated with a more thorough assessment. On the other hand, the application of Chilean criteria, although not compulsory, suggests Chilean soil conditions. Finally, this study reveals the need to count on background soil concentration values and also measurements of clay and organic materials for the tailings. These data will allow a clearer picture of the real risks associated with Chilean tailings.

## Figures and Tables

**Figure 1 ijerph-17-03948-f001:**
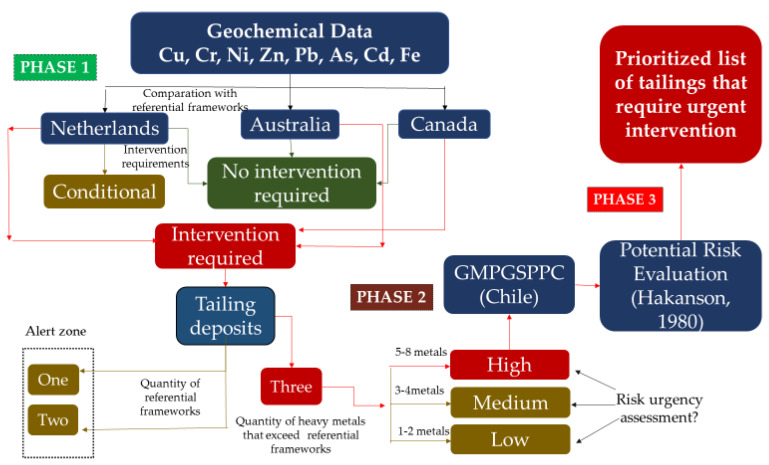
Description of the methodology.

**Figure 2 ijerph-17-03948-f002:**
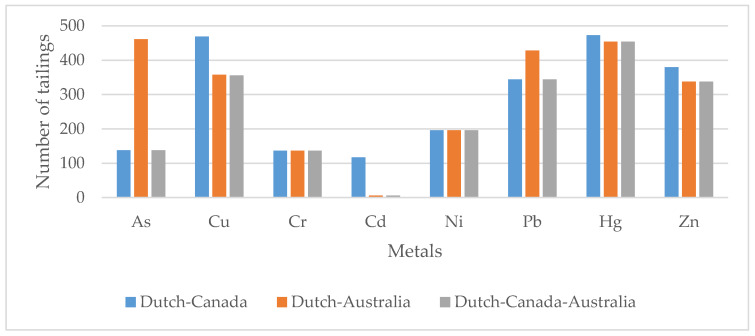
Comparison of results by applying the Canadian and Australian guidelines and the adapted Dutch norm to 530 tailings.

**Figure 3 ijerph-17-03948-f003:**
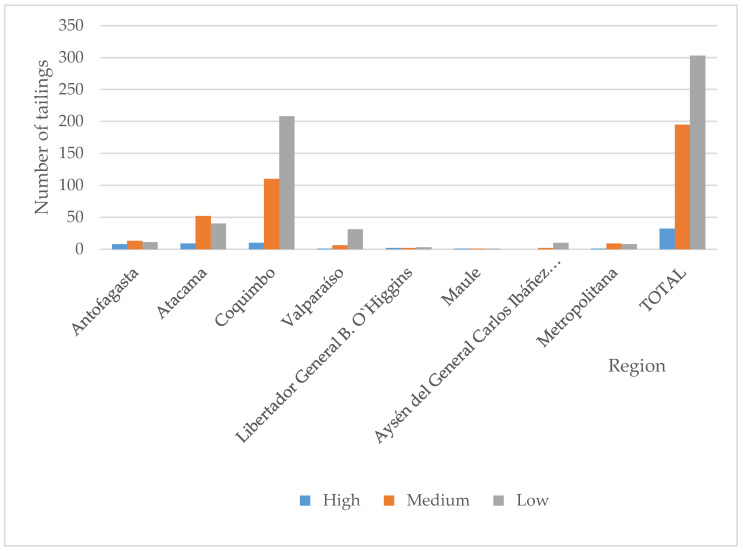
Priority classification of Chilean tailings by region.

**Figure 4 ijerph-17-03948-f004:**
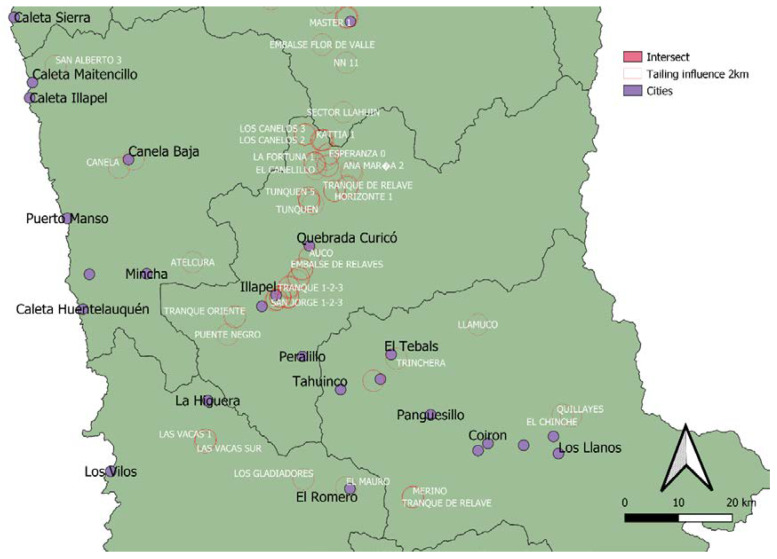
Map of the urban area intersections with tailings distribution in the Coquimbo region.

**Figure 5 ijerph-17-03948-f005:**
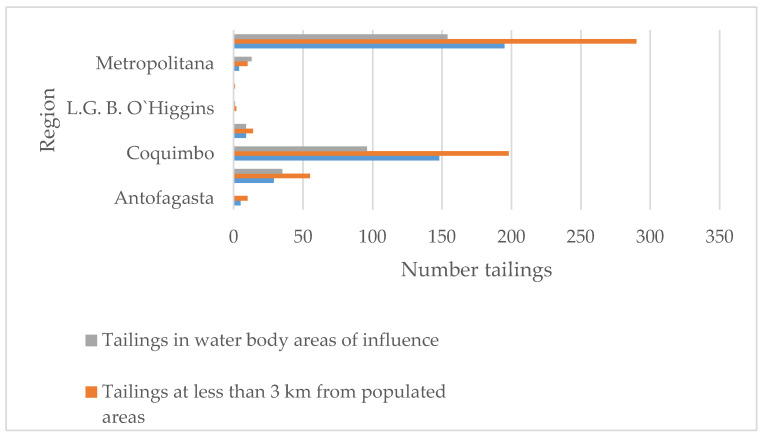
Distribution of tailings at less than 2–3 km from populates areas; tailings located close to water bodies (rivers, lakes, and estuaries).

**Figure 6 ijerph-17-03948-f006:**
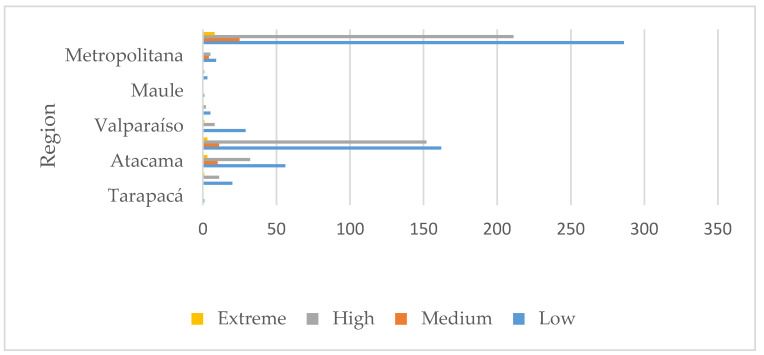
Regional distribution of tailings assessed according to heavy metals and closeness to populated areas.

**Figure 7 ijerph-17-03948-f007:**
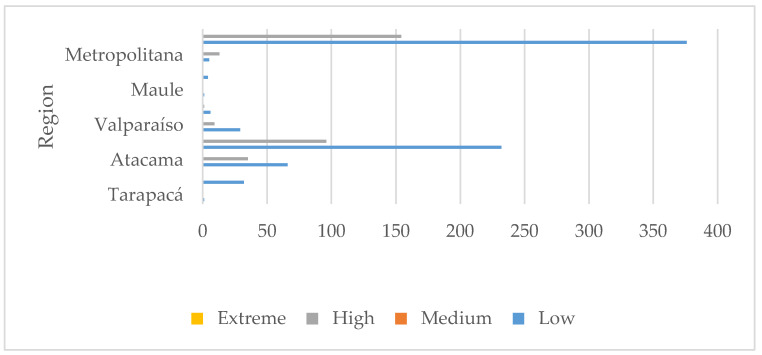
Distribution of regional tailings assessed according to heavy metals and water bodies.

**Figure 8 ijerph-17-03948-f008:**
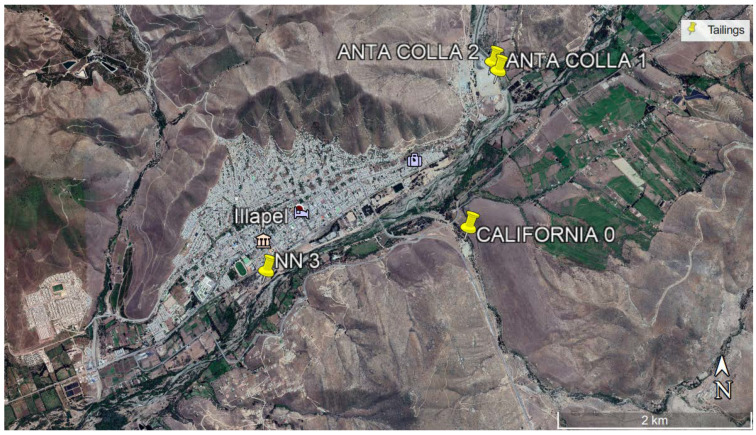
Tailings showing extremely high risk, according to Hakanson’s methodology [14].

**Table 1 ijerph-17-03948-t001:** Canadian [26] and Australian [27] soil quality reference values.

Parameter	Canada		Australia	
	Real Value ^1^	Parameterized Value ^1,2^	Real Value ^1^	Parameterized Value ^1,2^
	g ton^−1^			
As	12	1.1139	100	2.0043
Cd	10	1.0414	20	1.3222
Cr	64	1.8129	100	2.0043
Cu	63	1.8062	1000	3.0004
Hg	6.6	0.8808	15	1.2041
Ni	45	1.6628	600	2.7789
Pb	140	2.1492	300	2.4786
Zn	250	2.3997	7000	3.8452

^1^ Soils for residential use. ^2^ Parameterization with Equation (1).

**Table 2 ijerph-17-03948-t002:** Reference values for the SIV calculation of each element, according to the Dutch norm [28].

Element	A	B	C
As	15	0.4	0.4
Cd	0.4	0.007	0.021
Hg	0.2	0.0034	0.0017
Pb	50	1	1
Ni	10	1	0
Zn	50	3	1.5
Cu	15	0.6	0.6
Cr	50	2	0

**Table 3 ijerph-17-03948-t003:** Criteria for classifying tailings priority requirements.

Prioritized Intervention Requirements	Closeness to Communities			Metal Concentration over Reference Values ^1^			Closeness to Water Bodies	
	<2 km	2–3 km	>3 km	0–2	3–4	5–8	Yes	No
**High**								
**Medium**								
**Low**								

^1^ Applying the three international referential frameworks, red: requires hight priority, yellow: requires medium priority and green requires low priority.

**Table 4 ijerph-17-03948-t004:** Adjusted grading standard of potential ecological risk of heavy metals in soil [41].

$E_{r}^{i}$	Contamination Level	RI	Potential Ecological Risk
$E_{r}^{i}<30$	Low	RI < 40	Low
$30\;\leq E_{r}^{i}<60$	Middle	40≤RI<80	Middle
$60\;\leq E_{r}^{i}<120$	High	80≤RI<160	High
$120\;\leq E_{r}^{i}<240$	Very high	160≤RI<320	Very high
$240\;\leq E_{r}^{i}$	Extremely high	320≤RI	Extremely high

**Table 5 ijerph-17-03948-t005:** Results of the Dutch, Australian, and Canadian soil quality reference guidelines application to 530 Chilean tailings.

Reference	Requirements	As	Cu	Cr	Cd	Ni	Pb	Hg	Zn
Dutch norm	Intervention required	138	460	34	1	0	89	24	55
	No intervention required	323	11	107	116	196	372	449	325
	Conditional intervention	69	59	389	413	334	69	57	150
Canadian guidelines	Intervention required	0	521	238	37	324	255	29	181
	No intervention required	530	9	292	493	206	275	501	349
Australian guidelines	Intervention required	163	347	30	525	0	171	525	13
	No intervention required	367	183	500	5	530	359	5	517

**Table 6 ijerph-17-03948-t006:** Characteristics of the most critical tailings from a risk viewpoint, according to Dutch, Canadian, and Australian reference frameworks.

IDQ	Resource	Mass (t)	State	Region	Commune
1609	Cu	1,875,000	Inactive	Antofagasta	Sierra Gorda
1665	Cu	600	Abandoned	Antofagasta	Sierra Gorda
1639	Cu	14,080	Inactive	Antofagasta	Sierra Gorda
949	Cu-Au	40,005	Inactive	Coquimbo	Río Hurtado

**Table 7 ijerph-17-03948-t007:** Average heavy metal concentration values and potential ecological risk index for Illapel critical tailings.

Identification ^1^	Ecological Risk Factor $\;E_{r}^{i}$								Potential Ecological Risk Index
Name of tailing	Cu	Cr	Ni	Zn	Pb	As	Cd	Hg	RI
Anta Colla 2	53.4	5.6	9.2	0.7	460.4	1702.2	137.2	21.9	2391
Anta Colla 1	64.9	5.5	8.9	0.7	442.6	1607.9	33.3	106.7	2271
California 0	116.5	12.3	8.6	1.1	186.9	469.4	67.7	1280	2142
California 2B	11.5	6.9	7.5	0.5	24.1	40.0	135.8	24.8	251
El Arenal	42.3	5.4	7.9	2.5	60.1	40.0	77.8	48.6	285
Esperanza Dos	81.6	8.7	9.0	0.6	78.7	40.0	123.0	244.3	586
NN 3	26.3	5.2	7.5	0.6	69.1	40.0	132.7	445.7	727
Pluma de oro	7.3	9.7	7.9	0.7	26.6	40.0	80.7	69.5	242
San Jorge 1-2-3	31.8	7.3	6.7	0.9	22.6	40.0	123.3	32.4	265
Tailings dam 0	35.0	11.3	9.6	1.1	50.4	40.0	70.6	27.9	246
Tailings dam 1	45.3	5.6	7.1	1.0	31.5	40.0	103.3	40.0	274
Tailings dam 2	35.2	6.0	7.6	1.0	40.4	40.0	100.1	32.4	263

^1^ Identification according to National Geology and Mining Service (SERNAGEOMIN) [4].

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
