# Peer review of "Methodology to Prioritize Chilean Tailings Selection, According to Their Potential Risks"

_ijerph, 2020, doi:10.3390/ijerph17113948_

Round 1

Reviewer 1 Report

This work deals with economic, environmental, and social impacts caused by mining activity in Chile, more specifically with the prioritization of mine sites that need restoration. Although this is an important subject and it is within IJERPH scope, the outcomes of this work are not relevant to a broad swath of readers and the quality of presented data does not justify its publication in IJERPH. In addition, I am not convinced that the methodology used is able to take into account the complexity and diversity related to contaminated mine sites.

  1. Authors affiliation list: each institution should be listed only once and only the contact of the corresponding author should de provided.
  2. Overall, the manuscript does not flow and it is difficult to follow. The English is not sufficient for the peer review process.
  3. The whole manuscript must be carefully reviewed. It was poorly written and carelessly prepared: font size and theme font are not consistent through the text, message such as “Error! Reference Source Not Found.” Can be found in the manuscript, there are two figure 3, etc…
  4. Line 22-24: “One of the main challenges the country faces today is the 22 presence of more than 700 mine tailings containing eight EC environmental compounds for mining: 23 Cu, Cr, Ni, Zn, Pb, As, Cd, and Fe, among others.” Something is missing in the sentence in bold.
  5. Keywords: they are too vague and must be reviewed. “Environmental impact” and environmental risks” are redundant.
  6. Introduction is way to long. One page would be ideal. Some of its content may be moved to supporting material if needed. I would also suggest to review the order that information is presented. It does not flow.
  7. Line 57-58: The latter establishes that, once a mining activity ends, it must remain physically and chemically stable…” Please explain.
  8. Table for, last column: should be “parametrized” instead “oarametrized”.
  9. Resolution of equations is poor and needs to be improved.
  10. 21 tables are way too many for a manuscript. Authors must decide which ones are relevant and which ones may be moved to supporting material or even withdrawn.
  11. Table 12: should be “Canada” instead “Canadá”.
  12. Figure 2 and 3 (maps) are not readable.
  13. Conclusion: first paragraph is actually relevant to introduction. The conclusion is more likely a summary of methodology and results. It should focused on the outcomes of the work and future research recommendation.

Reviewer 2 Report

Dear authors

Your paper is overall fine but there are minor issues that need to be addressed. Most of them are very simple. They are all highlighted and commented on the pdf file of your manuscript (attached). Some issues that you do need to address:

  • The aim of the manuscript is not clear. You need to add one or two sentences in the introduction, explaining what you want to achieve and it is important.
  • Equation 1 is not correct. You have C (concentration value) in both sides of the equation. The left C is in fact the parametrized value, which is obviously different from the concentration value.
  • In equation 2, you did not define what is the variable SIV neither explained how it is obtained. This needs to be explained.
  • The maps are very poor. They do not have coordinates neither scale and the legend is in Spanish. Figure 3 is terrible.
  • There are two figures 3.
  • There are some paragraphs very hard to understand what you mean. You need to make the sentences shorter and, in some cases, explain better.

These are simple and easy things to fix.

Reviewer 3 Report

The manuscript entitled "Methodology to prioritize Chilean tailings selection, according to their potential risks" showed results of evaluating environmental risks by some harmful elements released from tailings, with using a numerical model. This manuscript has too complex but insufficient explanations in all (Introduction, Materials and Methods, and Results and Diccussion) sections, with unaccessible English-writing and some mistypes. The authors should subtilize the content of this manuscript and receive a writing check by an English-native scientist. Addttionally, I seem that this papaer is just like an 'administrative report', not a scientific paper. The authors must make an effort to show 'scientific' importances of their paper in the sections of Introduction and, Results and Discussion.   

Introduction

  1. The historical explanations for tailing in Chili are too long, while explanations for the methodological aproach are insufficient.
  2. The aim and final goal are not clearly shown.
  3. Descriptions for mehodology should be transfered to Materials and Methods.               

Materials  and Methods

  1. There are many unaccessible abbreviations in the text and tables, respectively. Additionally, please put units for values in some tables.
  2. There are too many tables. Parts of the tables will be discarded or combined.
  3. A map data such as location of mines and characteristics of soil should be shown by figures.

Results and Discussion

  1. There are too many tables, while the authos did not explain values in the tables in details. If they did not explain, the tables should be summerized or transfered to supporting information.
  2. There are few descriptions related to an adequacy of the methodology for risk assessment. I seem that the authors just only estimated and did not evaluated the performance of the present method.

Reviewer 4 Report

This paper is focused on a methodology to prioritize Chilean tailings selection, according to their potential risks. The subject of the manuscript falls within the scope of the journal and the results of the paper are presented accordingly. The analysis herein needs quite improvement. Interpretations are adequate, although conclusions deserve improvement. A broader discussion is expected and the results shown should have a comparison/contrast with others obtained elsewhere to broaden the spectrum of the survey and make the study interesting for a general audience; Organization of the article admits improvement in its structure to better deploy the study and presents its relevance which is not clearly perceived.

Specific points to be addressed are:

 - Novelty must be highlighted;

- References are adequate but in some form they might be improved in terms of number and relevance to bring a better picture of the problem and the way it can be generalized to a greater audience. When using references, because they are few and not exactly comprehensive they appear several times and in sentences like this “As a whole Chile uses the Dutch norms and data from the Canadian guidelines [5]” where the citation is just repeating the paragraph above and not justifying or validating that Chile uses Dutch and Canadian standards;

- The structure of the paper should be revised in a way that one can better understand the research and better follow its presentation. An introduction to the study separated from the literature review, a methods section, the case study presentation, results and discussion followed by conclusions is recommended. Figures and tables presented in a more organized way, with maybe, some of them put in an appendix is also adequate;

- An English review to improve the text in grammar, style and to avoid typos is recommended;

- A general framework in a graphical form or a flowchart is desirable. As the paper is now it is necessary to apply some effort to fully grasp the method and this better description would improve the paper a great deal. A full description of the method and its sequence will stress how one can replicate the study elsewhere (even other researchers in Chile);

- To allow the questions and findings here be used by a large community conclusions must be universal and become applicable to a larger set of places and situations, how is this possible? How can one apply these findings and be secured that this is not the best to way to do it? The paper needs a better description of the questions made and the hypotheses taken to allow full evaluation or understanding of the findings;

Round 2

Reviewer 1 Report

The authors made all suggested modifications and I am impressed with how the quality of the manuscript has significantly improved.

Two minor changes are still needed:

  • I suggest to increase font size in figures 1 and and 5; and
  • Table 3: operators >, <, etc are overlapping.

Reviewer 3 Report

Although the authors made some efforts to answer my comments, it was insuffcient. My additional comments was described in the follwing:

  1. Definitions for abbreviation are not added in some parts.
  2. Unnatural grammers are found out in many parts.
  3. Reference data for the introduction of line 48-59 are too little. This part seems to be just like author's image.
  4. Introduction or explanation for your methodology is insufficient. For example, why the authors selected Hakanson's method, and Dutch, Canadian, and Australian criteria? I think that they need to explain that their approch was considered to be best or valuable when they had started to study. Addtionally they had better explain characteristics of the foreign criteria.
  5. Discusstion should be described after the reason of criteria makes clear described in the above comment'4. 

Reviewer 4 Report

The document was highly improved. Minor things still can be recommended as the authors could revise the abstract to enhance it a little bit removing excessive numbers. Another point is the question of norm versus standard nomenclature that should be determined better, Other minor points are final touches that one sees in a final version of the manuscript.  
